# The association between physical activity and erectile dysfunction: A cross-sectional study in 20,789 Brazilian men

Rafael Mathias Pitta[1]*, Oskar Kaufmann[1], Andressa Cristina Sposato Louzada[1], Rafael Haddad Astolfi[1], Luana de Lima Queiroga[1], Raphael Mendes Ritti Dias[2], Nelson Wolosker[1]

1 Inst Israelita Ensino & Pesquisa, Postgrad Program Hlth Sci, São Paulo, SP, Brazil, 2 Univ Nove Julho, Postgrad Program Rehabil Sci, São Paulo, SP, Brazil

* prof.rafaelpitta@gmail.com

## Abstract

### Introduction

Erectile dysfunction, defined as the inability to achieve and/or maintain a penile erection sufficient for satisfactory sexual intercourse is associated with impaired quality of life and cardiovascular diseases in men older than 40 years.

### Objective

To evaluate the association between erectile dysfunction and physical activity levels in a large cohort of men.

### Methods

Data from 20,789 males aged 40 years and over who participated in the check-up screening between January of 2008 and December of 2018 were included in this study. In this sample, data about erectile dysfunction, physical activity levels, clinical profile and laboratory exams were obtained. Logistic regression models were performed.

### Results

Individuals with erectile dysfunction were older (49.1 ±6.9 vs. 54.8±8.8 years old, p<0.001), had a higher body mass index (27.6 ±3.9 vs. 28.5 ± 4.3 kg/m², p<0,001), and presented with a higher prevalence of physical inactivity (25 vs. 19%, p<0.001) than individuals without erectile dysfunction. The multivariate model revealed that age (p<0.001), hypertension (p = 0.001), diabetes mellitus (p<0.001), high body mass index (p<0.001), lower urinary tract symptoms and depressive symptoms (p<0.001) were independent risk factors for erectile dysfunction. Low or high physical activity levels (OR = 0.77; CI95%: 0.68–0.87, p<0.001 and OR = 0.85; CI95%: 0.72–0.99, p = 0.04 respectively) were protective factors against erectile dysfunction.

**Data Availability Statement:** All relevant data are within the paper and its Supporting Information files.

**Funding:** The author(s) received no specific funding for this work.

**Competing interests:** The authors have declared that no competing interests exist.

## Conclusion

Low and high physical activity levels were associated with more than 20% reduction in the risk of erectile dysfunction in men aged 40 years or older.

## Introduction

Erectile dysfunction [ED], defined as the inability to achieve and/or maintain a penile erection sufficient for satisfactory sexual intercourse [1], is a prevalent condition in men older than 40 years [2–6], reaching up to 70% in older adults [2, 4, 6]. In addition to being associated with the impairments in quality of life [1], ED has been associated with cardiovascular diseases [2, 7] and the risk of cardiovascular events [7].

Efforts should be made to identify and reduce ED risk factors, especially the modifiable ones [8]. In this regard, several studies have demonstrated the benefits of physical activity levels [PAL] as a protective factor for ED, including meta-analyses of randomized controlled trials [8] and real-world cross-sectional population studies [9]. The main advantage of targeting PAL as a strategy to prevent ED is that it was the only action that was proven to significantly reduce the risk of ED even in middle age [10] when ED is more frequently diagnosed. [10]

Meta-analyses of randomized controlled trials [8] have shown that physical exercise decreases ED and real-world cross-sectional population studies [9] have shown that physical activity levels (PALs) are associated with ED. However most cross-sectional studies were carried out with PALs and ED assessment instruments not validated by the literature and in small populations, different age groups and were not adjusted for confounding variables [9], which limits the multifactorial understanding of the relationship between ED and PAL.

As cultural and sociodemographic factors affect the perception and treatment-seeking of ED [4], knowing the local epidemiology of ED is paramount to planning prevention strategies. Furthermore, it is preferable to apply validated questionnaires to large populations to better design the epidemiological assessments.

To the best of our knowledge, only 3 small studies have investigated the epidemiology of ED in Brazil. Rhoden et al. [6] assessed ED in 965 men using the abridged 5-item version of the International Index of Erectile Function [IIEF-5] but studied only the prevalence of ED and its relationship with age. Both Nicolosi et al. [4] and Moreira Jr. [11] et al. studied ED and its association with various risk factors in 600 Brazilian men each. These studies used validated questionnaires to assess depressive symptoms and lower urinary tract symptoms [CES-D and IPSS, respectively] but not to assess ED or PAL, which were both evaluated with single generic questions [4, 11].

Therefore, we designed the present study to evaluate ED and its risk factors, with a particular focus on PAL, adjusting for confounding variables (clinical, laboratory, and behavioral) and applying validated questionnaires, such as the International Index of Erectile Function [IIEF-5] and the International Physical Activity Questionnaire [IPAQ], in a sample of up to 20.000 Brazilian adults.

## Materials and methods

### Design

The present study is a retrospective cross-sectional analysis with primary outcomes to verify the association between PAL and DE in adults. Health data were collected from a large cohort

of men aged 40 years old or older who participated in health screening initiatives in the Preventive Medicine Center at Hospital Israelita Albert Einstein between 2008 and 2018. The Ethics Committee of the Hospital Israelita Albert Einstein approved this study (CAAE 94867018.6.0000.0071). A waiver of informed consent was requested and granted.

## Participants and settings

Initially, data from 44,395 male check-ups were included in the database. In individuals with duplicate data, i.e., more than one preventive medical visit, we considered only the most recent visit. Than, we excluded male check-ups with missing data on PAL and ED, participants who reported no sexual activity in the last year as evaluated with a self-assessment binary question (positive or negative), and those with penile prostheses. Only 10 patients underwent radical prostatectomy and were not excluded. Finally, data on PAL and ED from 20,789 males over 40 years were analyzed.

## Clinical data

Age at the time of the preventive medicine visit was registered.

Height and weight were obtained to calculate body mass index [BMI] using an InBody 230 scale (Ottoboni®) and a stadiometer, respectively, with an accuracy of 0.1 mm. Waist circumferences were measured with tape with an accuracy of 0.1 cm.

Blood pressure was measured in triplicate according to the standard method recommended by the American Heart Association [12].

Comorbidities such as systemic arterial hypertension, diabetes mellitus, dyslipidemia, tobacco use, nonalcoholic fatty liver steatosis [NASH], and continuously used medications were reported by each patient or assessed through medical records when available.

Data regarding alcohol consumption, depressive symptoms, perceived stress, PAL and ED were assessed through face-to-face interviews by trained professionals using dedicated and validated questionnaires. Questionnaires used were the Alcohol Use Disorders Identification Test [AUDIT] [13], Beck Depressive Inventory [BDI] [14], Perceived Stress Scale [PSS] [15], IPAQ [16] [S1 Fig] and IIEF-5 [17].

For the analysis or PAL, IPAQ provides information on walking time, vigorous- and moderate-intensity activity and sedentary activity in a usual week. Individuals who engaged in at least 30 minutes of vigorous physical activity at least 5 days per week or those who engaged in at least 20 minutes of vigorous physical activity at least 3 days per week or associated with moderate physical activity and/or walking for at least 30 minutes on at least 5 days per week were classified as highly active. Individuals who practice at least 20 minutes of vigorous physical activity at least 3 days a week or those who practice any type of physical activity for at least 150 minutes a week spread out over at least 5 days were considered active. Individuals who reported engaging in physical activity but did not meet the criteria above, were classified as moderately active. Individuals who reported no physical activity were classified as sedentary.

## Laboratorial data

Blood samples were collected after an overnight fast and analyzed as part of a routine clinical workflow. Laboratory analyses included determination of glycosylated hemoglobin percentage (%), a standard lipid panel (mg/dL) and uric acid levels (mg/dL). The laboratory responsible for all blood analyses meets the standardized criteria for quality control established by the Brazilian Health Ministry.

## Data analysis

Our main objective was to evaluate the association between ED and PAL, controlled by previous clinical features.

When data were unavailable in medical records, we considered a patient to be hypertensive if they self-reported hypertension or if they self-reported continuous use of antihypertensive medication. Similarly, we considered a patient to have diabetes if they self-reported diabetes mellitus or if they self-reported continuous use of anti-diabetic medication. Metabolic syndrome was defined as recommended by the World Health Organization [18].

Missing data: We excluded all patients with missing data on ED and/or PAL, but not on other variables. Totals varied according to available data and were accordingly recorded.

Statistical analyses were conducted using SPSS for Windows Version 24.0 (IBM Corp, Armonk, NY, USA). Participant characteristics were presented using frequencies and percentages for categorical variables, while the means and standard deviations [SDs] were used for continuous variables. Data normality was analyzed using the Shapiro-Wilk test. In the comparison of categorical variables, the chi-square test was used. In the comparison of numerical variables, Student's t test and the Mann-Whitney test were used according to the normality of the data.

We used a convenience sample because it is a specific population study. However, a sample calculation was performed where considering a 0.20 lower odds ratio compared with the reference value, a power of 80% and an alpha error of 5% the sample size was estimated in 4233 subjects.

The crude associations between erectile dysfunction and physical activity or other characteristics were examined using odds ratios.

To run a logistic regression model, the IIEF-5 scores were categorized as follows: presence of erectile dysfunction ($\leq$21 points, including severe, moderate, mild to moderate and mild categories) and absence of erectile dysfunction (>21 points). A $p$ value < 0.05 was considered significant. Adjusted odds ratios (aOR) and 95% confidence intervals (95% CIs) were computed for the logistic model results.

## Results

### Demographic and anthropometric data

In total, we studied 20,789 men aged 40 to 91 years. The median age was 49 years old and most men were between 40 and 64 years old (95.44%).

Table 1 presents the comparison of demographic and clinical data of study participants in relation to ED. Individuals with ED were older (49.10±6.85 vs. 54.75±8.81 years old, p<0.001) and had higher BMI (27.62±3.93 vs. 28.49±4.30 kg/m$^2$, p<0.001) than individuals without ED.

**Table 1. Comparison of demographic and clinical variables in relation to ED (n = 20,789).**

| Variables | ED | Mean | SD | N | p-value |
|---|---|---|---|---|---|
| Age | Absence | 49.10 | 6.85 | 17447 | <**0.001** |
| | Presence | 54.75 | 8.81 | 3342 | |
| | Total | 50.06 | 7.52 | 20789 | |
| BMI (kg/m$^2$) | Absence | 27.62 | 3.93 | 17446 | <**0.001** |
| | Presence | 28.49 | 4.30 | 3343 | |
| | Total | 27.77 | 4.01 | 20789 | |

t-Student test, * Mann-Whitney test

ED: erectile dysfunction, SD: standard deviation, n: sample size, BMI: body mass index, kg/m$^2$: kilogram/ square meter.

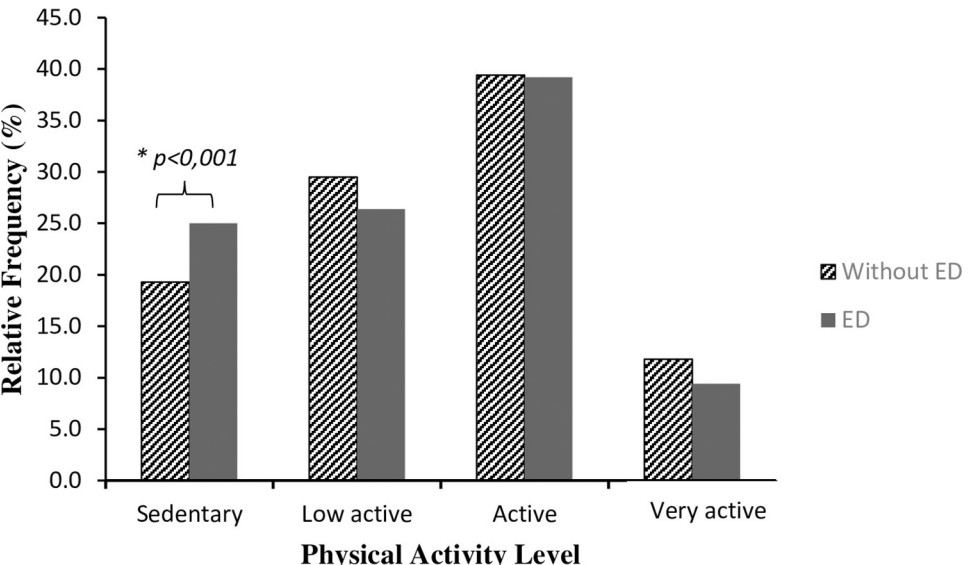

**Fig 1. Level of physical activity among men with and without erectile dysfunction (n = 20,789).**

## Erectile dysfunction and physical activity level

In total, 3,560 men (17.12% of the participants) reported ED. The participants were distributed in the IIFE-5 classes as follows: 82.88% had no ED, 11.58% had mild ED, 3.3% had mild to moderate ED, 1.13% had moderate ED, and 1.12% had severe ED.

The distribution of PAL in individuals with and without ED is shown in Fig 1. In total, 20.34% of the individuals were sedentary, 28.98% were low active, 39.37% were active and 11.37% were high active. Individuals with ED were significantly less active (p<0.001).

## Comorbidities

Patients with ED had significantly more comorbidities associated with higher cardiovascular risk, as shown in Table 2 (p<0.001 for all comparisons).

Patients with ED had a significantly higher prevalence of lower urinary tract symptoms (p<0.001), as shown in Table 3.

**Table 2. Relative frequencies (%) of comorbidities in relation to ED (n = 20,789).**

| Variable (n) | Relative frequency (%) | | | *p*-value |
| --- | --- | --- | --- | --- |
| | All patients | With ED | Without ED | |
| Hypertension (20,789) | 26.5 | 39.6 | 23.8 | <0.001 |
| Diabetes mellitus (20,789) | 8.4 | 16.1 | 6.8 | <0.001 |
| Dyslipidemia (20,789) | 53.9 | 57.1 | 53.2 | <0.001 |
| Use of hypolipemiant (20,789) | 21.1 | 27.8 | 19.7 | <0.001 |
| Metabolic syndrome (20,789) | 10.7 | 15.5 | 9.7 | <0.001 |
| *Non-alcoholic fatty liver steatosis* (20,789) | 51.9 | 58.1 | 50.7 | <0.001 |

Chi-squared test; n = sample size

**Table 3. Relative frequencies (%) of urological comorbidities in relation to ED (n = 20,789).**

| Variable | | Relative frequency (%) | | | p-value |
|---|---|---|---|---|---|
| | | All patients | With ED | Without ED | |
| Lower urinary tract symptoms (20,789) | Absent/mild | 91.8 | 78.9 | 94.5 | <0.001 |
| | Moderate | 7.3 | 18.3 | 5 | |
| | Severe | 0.9 | 2.8 | 0.5 | |

Chi-squared test.

## Laboratory data

When analyzing laboratory test results associated with increased cardiovascular risk, we observed that glycosylated hemoglobin was significantly higher in men with ED (5.79 ± 1.01 vs. 5.57 ± 0.67, p<0.001), uric acid levels were not different (p = 0.658), and lipid profile results were conflicting. Regarding lipid profiles, triglycerides (155.27 ± 119.45 vs. 149.17 ± 108.28, p = 0.002) and high-density lipids (44.37 ± 11.08 vs. 45.72 ± 11.16, p<0.001) were worse in men with ED, but low-density lipids (114.53 ± 34.81 vs. 122.25 ± 34.01, p<0.001) were better. The laboratory test results are summarized in Table 4.

## Behavioral assessment

Psychological data and lifestyle habits are shown in Table 5. Patients with ED had a higher prevalence of tobacco use, consumed alcohol at higher levels, and exhibited depressive symptoms but had a lower prevalence of perceived stress (p<0.001 for all comparisons).

## Predictors of erectile dysfunction

After performing a full multiple logistic regression model (S1 Table), we performed a stepwise backward multiple logistic regression, analyzing only the variables associated with statistically significant risk or protection factors for ED. In this final model, age, hypertension, diabetes mellitus, BMI, lower urinary tract symptoms, and depressive symptoms were strong independent risk factors for ED. At the same time, nonsmoker status and low or high PAL were strong independent protective factors, as shown in Table 6.

**Table 4. Laboratory test results in relation to ED (n = 20,789).**

| Variable (n) | Mean ± SD | | | p-value |
|---|---|---|---|---|
| | All patients | With ED | Without ED | |
| TC (20,789) | 194.87 ± 39.00 | 187.52 ± 40.42 | 196.37 ± 38.53 | <0.001 |
| HDL (20,789) | 45.49 ± 11.16 | 44.37 ± 11.08 | 45.72 ± 11.16 | <0.001 |
| LDL (20,789) | 120.94 ± 34.27 | 114.53 ± 34.81 | 122.25 ± 34.01 | <0.001 |
| TG (20,789) | 150.20 ± 110.27 | 155.27 ± 119.45 | 149.17 ± 108.28 | 0.002 |
| UA (20,789) | 5.99 ±1.38 | 5.98 ± 1.52 | 6.00 ±1.35 | 0.658 |
| HbA1c (20,789) | 5.61 ± 0.75 | 5.79 ± 1.01 | 5.57 ± 0.67 | <0.001 |

t-Student test; n = sample size; TC: total cholesterol, HDL: high-density lipids, LDL: low-density lipids, TG: triglycerides, UA: uric acid, HbA1c: glycosylated hemoglobin

**Table 5. Relative frequencies of tobacco use, risky alcohol consumption, perceived stress, depressive symptoms, and lower urinary tract symptoms in relation to ED (n = 20,789).**

| Variable (n) | | Relative frequency (%) | | | *p*-value |
|---|---|---|---|---|---|
| | | All patients | With ED | Without ED | |
| Tobacco use (20,789) | Never | 70.1 | 63.9 | 71.4 | <0.001 |
| | Previous | 20.7 | 26.4 | 19.5 | |
| | Active | 9.2 | 9.7 | 9.1 | |
| Alcohol consumption (20,789) | Low-risk | 83.5 | 81.3 | 84 | <0.001 |
| | Hazardous | 14.4 | 16 | 14.1 | |
| | Moderate-severe alcohol use disorder | 2.1 | 2.7 | 1.9 | |
| Perceived stress (20,789) | Absent | 79.5 | 74.5 | 80.6 | <0.001 |
| | Present | 20.5 | 25.5 | 19.4 | |
| Depressive symptoms (20,712) | Absent | 86.3 | 76.5 | 88.3 | <0.001 |
| | Present | 13.7 | 23.5 | 11.7 | |

Chi-squared test; n = sample size

## Discussion

This study provides the largest evaluation of regional prevalence and predicting factors of ED ever conducted in Brazil. It is the only Brazilian study using validated questionnaires to assess both ED and PAL. We identified that the physical activity level was associated with a lower risk of ED, even when controlling for other risk factors.

The high prevalence of ED is even more relevant to the health care system as this disease has not only been shown to significantly impair the individual's quality of life [19] and inter-personal relationships [20] but is also a predictor of cardiovascular disease [2, 7]. Our findings

**Table 6. Predictors of ED (n = 20,789).**

| Variable | OR | CI (95%) | | p |
|---|---|---|---|---|
| Age | 1.084 | 1.078 | 1.091 | <0.001 |
| Hypertension | 1.186 | 1.077 | 1.307 | 0.001 |
| Diabetes mellitus | 1.364 | 1.192 | 1.561 | <0.001 |
| Body mass index | 1.029 | 1.018 | 1.040 | <0.001 |
| Tobacco use | | | | |
| *Previous* | 0.897 | 0.811 | 0.992 | 0.035 |
| *Active* | 1.144 | 0.991 | 1.320 | 0.066 |
| Physical Activity Level | | | | |
| *Low active* | 0.771 | 0.685 | 0.868 | <0.001 |
| *Moderate* | 0.912 | 0.816 | 1.019 | 0.103 |
| *High Active* | 0.849 | 0.724 | 0.995 | 0.044 |
| Lower urinary tract symptoms | | | | |
| *Moderate* | 2.764 | 2.440 | 3.132 | <0.001 |
| *Severe* | 3.133 | 2.257 | 4.349 | <0.001 |
| Depressive symptoms | 2.212 | 1.992 | 2.457 | <0.001 |
| HDL | 0.994 | 0.990 | 0.998 | 0.002 |
| LDL | 0.998 | 0.996 | 0.999 | 0.001 |

Multiple logistic regression (Stepwise backward)

corroborate the association between ED and atherosclerosis reported by other authors [2, 4], as we observed that independent risk factors for ED were conditions that are also well-established risk factors for atherosclerosis, such as age, systemic arterial hypertension, diabetes mellitus, and obesity [21].

Aging is one of the most important independent factors responsible for increasing the prevalence and severity of ED, as shown in numerous studies [2, 3–6] and as observed in our analysis. The association between ED and aging is most likely related to the process of atherosclerosis, leading to penile vascular arteriopathy, that limits arterial blood flow, and to symptomatic coronary artery disease that limits sexual physical performance [22]. In addition, aging is also associated with decreased testosterone levels, and impaired libido, sexual function, physical fitness, and mood [23]. Nevertheless, although ED increases with age, it is not an inevitable outcome of the aging process. Therefore, efforts should focus on controlling comorbidities and mainly on the promotion of healthy lifestyle habits.

Among the comorbidities that are risk factors for atherosclerosis and ED, having DM was the strongest predictor for ED in our study. Diabetic men are reported to have an earlier onset of ED that presents with greater severity and poorer response to its treatment [24]. This is probably associated with synergistic vascular, neurological, and endocrine abnormalities, including an association with low levels of testosterone [25].

On the other hand, our findings regarding the association between ED and dyslipidemia were conflicting, as higher HDL levels were an independent protective factor; however, so was high LDL levels, although their protective effect had little expression. Other authors have also reported different findings regarding the association of ED and dyslipidemia, even when studying men from the same country. Pinnock et al. [26] reported that a high cholesterol level was an independent predictor of impotence in their study of 612 men in Australia. In contrast, Weber et al. [5] observed no such association when studying over one hundred thousand Australian men. In fact, most studies that reported dyslipidemia as a risk factor for ED used total cholesterol as measure, while we separately analyzed the components of the lipid profile. Further studies are necessary to understand the role of each type of cholesterol in ED.

Having depressive symptoms doubled the risks of presenting with ED, in line with other authors' reports [3, 4]. Indeed depression and ED are frequently correlated and likely feed back into each other [19, 27]. Additionally many anti-depressive drugs lead to ED, and patients with ED and depressive symptoms are more likely to discontinue treatment for ED [19]. Therefore, special attention should be given to this association when planning and assessing its treatments.

Finally, another known strong independent factor associated with ED that was also likewise observed in our study was clinically significant lower urinary tract symptoms (LUTS) [28]. Since most medical treatments for LUTS can lead to negative sexual symptoms, ED should be actively evaluated when monitoring and selecting the most appropriate treatment.

One of the most important aspects of the therapeutic management of ED, along with pharmacological therapy, is identifying and treating reversible risk factors. Regarding lifestyle habits, quitting smoking and being physically active, even at low levels, were strong protective factors observed in our study. In the past decade, several studies have associated the practice of PAL with significant improvements in overall cardiovascular health and erectile function scores [8, 29]. Moreover, PAL may be one of the only truly effective measures to improve erectile function even when started in middle age, as there is little evidence that quitting smoking, reducing alcohol consumption, and losing weight can reverse the symptoms of erectile dysfunction [9].

The positive effect that PAL [30] has on improving sexual function is based not only on controlling classic cardiovascular risk factors, such as weight and cholesterol levels but also on

increasing the systemic bioavailability of endothelial-derived nitric oxide (NO), improving insulin sensitivity, which is also an important vascular NO release stimulator, lowering serum proinflammatory cytokines levels and increasing testosterone levels. Our results also support these findings since the multiple logistic regression model identified low and high PAL as independent protective factors for ED and was thus unrelated to age or other risk factors. When analyzing IPAQ classes separately, moderate PAL was not a significant protective factor, which may be due to confounding factors not addressed in our multivariate regression, such as testosterone replacement therapy and healthy eating habits. Nevertheless, the role of PAL as a protective factor against ED has been clearly demonstrated, and regular PAL, even at low levels, should be encouraged for men of all ages as an effective method to improve sexual function and cardiovascular health.

There are several potential limitations in our study. First, as previously mentioned, is the lack of information about the use of pharmacological treatment for ED, which could have reduced the prevalence of ED and influenced the impact of risk and protective factors on the disease. Second, the presence of a stable relationship for all included patients included in the study was not confirmed, which could be important to evaluate sexual function. Sexual orientation was also not collected.

In addition, men aged <40 years were excluded from our study. There is also the possibility that the study results cannot be generalized to the entire Brazilian population, since selection bias could not be prevented as the study sample included only men with private insurance and who participated in health check-ups.

On the other hand, our study also has several strengths. This study is the largest analysis of ED performed in South America. In addition, all participants underwent a detailed health examination by a physician, with a standardized assessment of medical history, physical examination, and laboratory exams. Furthermore, internationally validated questionnaires were utilized to assess the prevalence and severity of ED and LUTS, as well as to stratify PAL, strengthening the reliability and validity of the results and associations. We hope that the demonstrated results can be used in clinical and public health settings to positively influence individuals in physical activity adherence and erectile dysfunction prevention strategies, since men who are taking better care of their overall health (e.g. by avoiding health-risk behaviors) are also those who follow more physical lifestyle. In this way physical exercise can, to an extent, be a "surrogate marker" of better lifestyles.

## Conclusions

Our findings show that PAL is a strong independent protective factor against ED, even at low levels, regardless of age and comorbidities, and therefore should be strongly encouraged during the treatment and prevention of this condition.

## Supporting information

**S1 Fig. International physical activity questionnaire—short form.**
(PDF)

**S1 Table. Predictors of ED (n = 20,789).**
(DOCX)

**S1 Appendix. Predictors of ED (n = 20,789).**
(DOCX)

## Author Contributions

**Conceptualization:** Rafael Mathias Pitta, Oskar Kaufmann, Andressa Cristina Sposato Louzada, Raphael Mendes Ritti Dias, Nelson Wolosker.

**Data curation:** Rafael Mathias Pitta, Oskar Kaufmann, Luana de Lima Queiroga, Raphael Mendes Ritti Dias, Nelson Wolosker.

**Formal analysis:** Rafael Mathias Pitta, Oskar Kaufmann, Andressa Cristina Sposato Louzada, Luana de Lima Queiroga, Raphael Mendes Ritti Dias, Nelson Wolosker.

**Investigation:** Rafael Mathias Pitta, Luana de Lima Queiroga, Nelson Wolosker.

**Methodology:** Rafael Mathias Pitta, Rafael Haddad Astolfi, Luana de Lima Queiroga, Nelson Wolosker.

**Project administration:** Rafael Mathias Pitta, Nelson Wolosker.

**Resources:** Rafael Mathias Pitta.

**Software:** Rafael Mathias Pitta.

**Supervision:** Rafael Mathias Pitta.

**Validation:** Rafael Mathias Pitta.

**Visualization:** Rafael Mathias Pitta.

**Writing – original draft:** Rafael Mathias Pitta, Rafael Haddad Astolfi.

**Writing – review & editing:** Rafael Mathias Pitta, Rafael Haddad Astolfi.

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
