## [Decision Letter · Decision Letter 0]

6 Sep 2022

PONE-D-22-22679The association between physical activity and erectile dysfunction: a cross-sectional study in 20,789 Brazilian men.PLOS ONE

Dear Dr. Pitta,

Thank you for submitting your manuscript to PLOS ONE. After careful consideration, we feel that it has merit but does not fully meet PLOS ONE’s publication criteria as it currently stands. Therefore, we invite you to submit a revised version of the manuscript that addresses the points raised during the review process.

We look forward to receiving your revised manuscript.

Kind regards,

Celeste Manfredi

Academic Editor

PLOS ONE

Journal Requirements:

Additional Editor Comments:

Dear authors, in addition to comments of peer-reviewers, here other suggestions.

The English of whole paper should be revised by a native-speaker

Several sentences have no reference.

The meaning of all acronyms should be added when reported the first time in the text; besides, it should be added in the legends of tables and figures

-Abstract -

“is” prevalent, “is” associated

I suggest to remove this useless sentence from the abstract: “Statistical analyses were conducted using SPSS for Windows Version 24.0 (IBM Corp, Armonk, NY, USA)”.

I suggest to add all specifies p-values in this sentence of the abstract: “The age, hypertension, diabetes mellitus, high body mass index,lower urinary tract symptoms and depressive symptoms”

-Introduction-

The authors should replace “Despite the impairments“ with “In addition to being associated with the impairments”

I suggest to include the acronyms when used the first time between brackets

“… when ED is more frequently diagnosed” Lack of reference

“most cross-sectional studies were carried out with PAL and ED assessment instruments not validated by literature, in small populations, different age group and not adjusted for confounding variables” Lack of reference

-Methods-

How the patients' sexual activity was assessed (inclusion criterion)

Was the presence of a stable relationship for all included patients confirmed? It is important to evaluate sexual function. If not, the authors should add this limitation

Any information on sexual orientation of included patients?

Were blood sugar and testosterone not evaluated? They are part of the routine examinations for patients with ED

The authors should report the primary outcome evaluated. IIEF-5 or IPAQ?

Was a calculation of the necessary sample size performed? The author should add it

The authors should define low-intermedium-high activity.

-Results-

I suggest to review all tables. In several tables, sample size for some characteristics was higher than the total number of patients enrolled (20,789). If it is not an error, the authors should explain better why.

-Discussion-

The authors should underline in the limitations that men < 40 years were excluded (bias)

Reviewers' comments:

Reviewer's Responses to Questions

**Comments to the Author**

1. Is the manuscript technically sound, and do the data support the conclusions?

Reviewer #1: Partly

Reviewer #2: Yes

2. Has the statistical analysis been performed appropriately and rigorously? 

Reviewer #1: I Don't Know

Reviewer #2: Yes

3. Have the authors made all data underlying the findings in their manuscript fully available?

Reviewer #1: No

Reviewer #2: Yes

4. Is the manuscript presented in an intelligible fashion and written in standard English?

Reviewer #1: Yes

Reviewer #2: Yes

5. Review Comments to the Author

Reviewer #1: The manuscript describes an interesting study analyzing the association between physical activity and erectile dysfunction together with related risk factors. The study includes a large population of men. However, the manuscript lacks most of key results (tables etc.) which hampers to provide an adequate review.

1. In any case, methodology should be better explained. For instance, the IPAQ questionnaire should be better described. This is the key point in the study and description of scores and cut-off points for categories should be described. Statistical analyses and models of adjustment for logistric regressions should be described in Data Analysis section

2. In this sense, it should be interesting if IPAQ score relationships could be analyzed as a continuous variable.

3. Since the main results were not provided, I cannot determine their relevance. However, it is clear that Figure 1 is not the best way of presenting the results since the appraisal is not clear. In fact what does relative frequency mean? It is not explained in figure legends.

4. It is not clear if the intermediate PAL is associated or not with lower ED prevalence. This is completely disregarded in the Discussion.

Reviewer #2: Dear Authors, I have read your manuscript with interest (although a bit late). I have no major concerns with your paper.

Just a few comments:

1 - revise "hih active" and "erectyle" in figure 1

2 - it might be worth discussing the postential association between endurance exercise and hypogonadism in the so-called "exercise-induced hypogonadism" (recent publication, not authored by me: https://pubmed.ncbi.nlm.nih.gov/32082255/).

3 - it can also be argued that men who are taking better care of their overall health (e.g. by avoiding health-risk behaviors) are also those who follow a more physical lifestyle. Therefore, it might be interesting to mention that physical exercise can to an extent be a "surrogate marker" of better lifestyles.

6. PLOS authors have the option to publish the peer review history of their article (what does this mean?). If published, this will include your full peer review and any attached files.

Reviewer #1: No

Reviewer #2: No

---

## [Author Response · Author response to Decision Letter 0]

15 Oct 2022

First, thank you for reading and submitting your comments on my manuscript. Below are the answers to the requesteded questions.

R: Thanks for the consideration. This document was uploaded.

R: The document was placed in the requested location.

R: The document was placed in the requested location with the proper nomination.

R: The above items are not applicable to the submitted work.

We look forward to receiving your revised manuscript.

Kind regards,

Celeste Manfredi

Academic Editor

PLOS ONE

Journal Requirements:

We note that you have stated that you will provide repository information for your data at acceptance. Should your manuscript be accepted for publication, we will hold it until you provide the relevant accession numbers or DOIs necessary to access your data. If you wish to make changes to your Data Availability statement, please describe these changes in your cover letter and we will update your Data Availability statement to reflect the information you provide.

R: We will provide the repository information for our data upon acceptance.

Please include captions for your Supporting Information files at the end of your manuscript, and update any in-text citations to match accordingly. Please see our Supporting Information guidelines for more information: http://journals.plos.org/plosone/s/supporting-information. 

R: We have included captions for the Supporting Information files at the end of our manuscript and updated the citations in the text.

Additional Editor Comments:

Dear authors, in addition to comments of peer-reviewers, here other suggestions.

The English of whole paper should be revised by a native-speaker

R: The English of the entire article was revisited by a native-speaker – AMERICAN JOURNAL OF EXPERTS.

Several sentences have no reference.

R: Thank you for the consideration. The appropriate sentences were referenced.

The meaning of all acronyms should be added when reported the first time in the text; besides, it should be added in the legends of tables and figures

R: Thanks for the consideration. The meaning of all acronyms were added when reported the first time in the text, in the legends of tables and figures.

Abstract

“is” prevalent, “is” associated

R: We added the verb “is” in both words. 

I suggest to remove this useless sentence from the abstract: “Statistical analyses were conducted using SPSS for Windows Version 24.0 (IBM Corp, Armonk, NY, USA)”.

R: The sentence was removed from abstract.

I suggest to add all specifies p-values in this sentence of the abstract: “The age, hypertension, diabetes mellitus, high body mass index,lower urinary tract symptoms and depressive symptoms”

R: The p-values of the variables listed above were duly placed in the text.

Introduction

The authors should replace “Despite the impairments“ with “In addition to being associated with the impairments”

R: We made the requested change. 

I suggest to include the acronyms when used the first time between brackets

R: We included the acronyms when used the first time. 

“… when ED is more frequently diagnosed” Lack of reference

R: The reference has been placed as suggested.

“most cross-sectional studies were carried out with PAL and ED assessment instruments not validated by literature, in small populations, different age group and not adjusted for confounding variables” Lack of reference

R: The reference has been placed as suggested.

Methods

How the patients' sexual activity was assessed (inclusion criterion).

R: Sexual activity in the last year was performed by self-assessment of a binary question, which could be positive or negative. This complement was included in the text.

Was the presence of a stable relationship for all included patients confirmed? It is important to evaluate sexual function. If not, the authors should add this limitation.

R: As we did not confirm the presence or a stable relationship for all included patients, it was add in the limitations that: the presence of a stable relationship for all included patients was not confirmed, what could be important to evaluate sexual function. 

Any information on sexual orientation of included patients?

R: Sexual orientation was not studied. 

Were blood sugar and testosterone not evaluated? They are part of the routine examinations for patients with ED

R: We used glycosylated hemoglobin as marker in the evaluation of a health routine because it presents a “history” of glycemic behavior in recent months. 

We agree that Testosterone is an important factor in the relationship with erectile dysfunction but it was not evaluated in this study.

The authors should report the primary outcome evaluated. IIEF-5 or IPAQ?

R: The main outcome was IIEF, que foi associado ao IPAQ.

Was a calculation of the necessary sample size performed? The author should add it

R: 

We used a convenience sample because it is a specific population study. However, a sample calculation was performed in which considering a 0.20 lower odds ratio compared with reference value, a power of 80% and a alpha error of 5% the sample size was estimated in 4233 subjects. 

This was added in text.

The authors should define low-intermedium-high activity.

R: Physical activity categories were described in a paragraph in the text, just before the Laboratory date:

IPAQ provides information on walking time, vigorous- and moderate-intensity activity and sedentary activity in a usual week. Individuals who engage in at least 30 minutes of vigorous physical activity at least 5 days per week or those who engage in at least 20 minutes of vigorous physical activity at least 3 days per week associated with moderate physical activity and/or walking for at least 30 minutes on at least 5 days per week were classified as highly active. Individuals who practice at least 20 minutes of vigorous physical activity at least 3 days a week or those who practice any type of physical activity for at least 150 minutes a week spread over at least 5 days were considered active. Individuals who reported engaging in physical activity but did not meet the criteria above, were classified as moderately active. Individuals who reported no physical activity were classified as sedentary. 

Results

I suggest to review all tables. In several tables, sample size for some characteristics was higher than the total number of patients enrolled (20,789). If it is not an error, the authors should explain better why.

R: It was just an error. We updated data in the tables and figures.

Discussion

The authors should underline in the limitations that men < 40 years were excluded (bias)

R: We underlined in the limitations that men under 40 were excluded. I appreciate the contribution.

Reviewers' comments:

5. Review Comments to the Author

Reviewer #1: The manuscript describes an interesting study analyzing the association between physical activity and erectile dysfunction together with related risk factors. The study includes a large population of men. However, the manuscript lacks most of key results (tables etc.) which hampers to provide an adequate review.

1. In any case, methodology should be better explained. For instance, the IPAQ questionnaire should be better described. J. Statistical analyses and models of adjustment for logistic regressions should be described in Data Analysis section.

R: 

R: Below we describe IPAQ categories in more detail. This was also added to the manuscript.

IPAQ provides information on walking time, vigorous- and moderate-intensity activity and sedentary activity in a usual week. Individuals who engage in at least 30 minutes of vigorous physical activity at least 5 days per week or those who engage in at least 20 minutes of vigorous physical activity at least 3 days per week associated with moderate physical activity and/or walking for at least 30 minutes on at least 5 days per week were classified as highly active. Individuals who practice at least 20 minutes of vigorous physical activity at least 3 days a week or those who practice any type of physical activity for at least 150 minutes a week spread over at least 5 days were considered active. Individuals who reported engaging in physical activity but did not meet the criteria above, were classified as moderately active. Individuals who reported no physical activity were classified as sedentary. 

2. In this sense, it should be interesting if IPAQ score relationships could be analyzed as a continuous variable.

R: 

R: We analyzed IPAQ scores as categorical data, according to its categories. 

3. Since the main results were not provided, I cannot determine their relevance. However, it is clear that Figure 1 is not the best way of presenting the results since the appraisal is not clear. In fact what does relative frequency mean? It is not explained in figure legends.

R: Figure 1 was corrected.

4. It is not clear if the intermediate PAL is associated or not with lower ED prevalence. This is completely disregarded in the Discussion.

R: 

R: In the discussion, we stated:

“… the multiple logistic regression model identified low and high levels of physical activity as independent protective factors for erectile dysfunction, thus unrelated to age or to other risk factors. When analyzing IPAQ classes separately, moderate level of physical activity was not a significant protective factor, which may be due to confounding factors not addressed in our multivariate regression, such as testosterone replacement therapy, healthy eating habits, among others. Nevertheless, the role of physical activity as a protective factor against erectile dysfunction has been clearly demonstrated, and regular physical exercise, even at low levels, should be encouraged for men of all ages as an effective method to improve sexual function and cardiovascular health.”

Reviewer #2: Dear Authors, I have read your manuscript with interest (although a bit late). I have no major concerns with your paper.

Just a few comments:

R: First, I would like to thank you for reading and reviewing the document in question.

1 - revise "hih active" and "erectyle" in figure 1

R: The change has been made.

2 - it might be worth discussing the postential association between endurance exercise and hypogonadism in the so-called "exercise-induced hypogonadism" (recent publication, not authored by me: https://pubmed.ncbi.nlm.nih.gov/32082255/).

R: We appreciate the suggestion, but the physical activity questionnaire (IPAQ) does not classify the type of physical activity performed, but the volume and intensity of the activity practiced, whether endurance or resistance training. In addition, our work did not evaluate the testosterone data in the population, as this procedure is not commonly accessible to the ongoing health review in question. However, I believe that this suggestion serves as a basis for new discussions and works in the future. Thank you one more time.

3 - it can also be argued that men who are taking better care of their overall health (e.g. by avoiding health-risk behaviors) are also those who follow a more physical lifestyle. Therefore, it might be interesting to mention that physical exercise can to an extent be a "surrogate marker" of better lifestyles.

R: 

We appreciate your idea and inserted the following sentence in the last paragraph of the discussion: since men who are taking better care of their overal health (e.g. by avoiding health-risk behaviors) are also those who follow more physical lifestyle. This way physical exercise can to an extent be a “surrogate marker” of better lifestyles.

---

## [Editor Report · Decision Letter 1]

18 Oct 2022

The association between physical activity and erectile dysfunction: a cross-sectional study in 20,789 Brazilian men.

PONE-D-22-22679R1

Dear Dr. Pitta,

We’re pleased to inform you that your manuscript has been judged scientifically suitable for publication and will be formally accepted for publication once it meets all outstanding technical requirements.

Kind regards,

Celeste Manfredi

Academic Editor

PLOS ONE

Additional Editor Comments (optional):

None

---

## [Editor Report · Acceptance letter]

25 Oct 2022

PONE-D-22-22679R1 

The association between physical activity and erectile dysfunction: a cross-sectional study in 20,789 Brazilian men. 

Dear Dr. Pitta:

I'm pleased to inform you that your manuscript has been deemed suitable for publication in PLOS ONE. Congratulations! Your manuscript is now with our production department. 

Kind regards, 

on behalf of

Dr. Celeste Manfredi 

Academic Editor

PLOS ONE